# Molecular Re-Diagnosis with Whole-Exome Sequencing Increases the Diagnostic Yield in Patients with Non-Syndromic Retinitis Pigmentosa

**DOI:** 10.3390/diagnostics13040730

**Published:** 2023-02-14

**Authors:** Anna Wawrocka, Magdalena Socha, Joanna Walczak-Sztulpa, Grzegorz Koczyk, Anna Skorczyk-Werner, Maciej R. Krawczyński

**Affiliations:** 1Department of Medical Genetics, Poznan University of Medical Sciences, 60-806 Poznan, Poland; 2Biometry and Bioinformatics Team, Institute of Plant Genetics, Polish Academy of Sciences, 60-479 Poznan, Poland; 3Centers for Medical Genetics GENESIS, 60-529 Poznan, Poland

**Keywords:** retinitis pigmentosa (RP), targeted NGS, whole-exome sequencing, molecular re-diagnosis

## Abstract

Retinitis pigmentosa (RP) is a clinically and genetically heterogeneous group of disorders with progressive loss of photoreceptor and pigment epithelial function. Nineteen unrelated Polish probands clinically diagnosed with nonsyndromic RP were recruited to this study. We used whole-exome sequencing (WES) to identify potential pathogenic gene variants in molecularly undiagnosed RP patients, as a molecular re-diagnosis after having performed targeted NGS in the past. Targeted NGS allowed for identification of the molecular background in only 5 out of 19 patients. Fourteen patients who remained unsolved despite the targeted NGS were subjected to WES. WES revealed potentially causative variants in RP-related genes in another 12 patients. Together, NGS methods revealed the coexistence of causal variants affecting distinct RP genes in 17 out of 19 RP families, with a very high efficiency of 89%. With the improvement of NGS methods, including higher sequencing depth, broader target enrichment, and better bioinformatic analysis capabilities, the ratio of identified causal gene variants has significantly increased. Therefore, it is important to consider repeating high-throughput sequencing analysis in those patients in whom the previously performed NGS did not reveal any pathogenic variants. The study confirmed the efficiency and clinical utility of re-diagnosis with WES in molecularly undiagnosed RP patients.

## 1. Introduction

Retinitis pigmentosa (RP, MIM: 268,000) is a progressive and degenerative inherited retinal disorder that affects 1 in 4000 people worldwide. RP is classified as nonsyndromic; not affecting other organs or tissues; and syndromic with other neurosensory disorders, developmental abnormalities, and complex clinical phenotypes.

RP is characterized by an initial loss of rod function, resulting in defective dark adaptation and progressive loss of peripheral vision, often progressing into the central retina involving cone photoreceptors and leading to severe visual impairment or blindness. Furthermore, characteristic “bone spicules” pigment depositions in the retina, attenuation of the retinal blood vessels, and optic disc pallor are observed [1,2,3]. Phenotypic variability among individuals sharing the same genetic background is observed. Moreover, RP is a genetically heterogeneous disease and can be inherited in an autosomal dominant, autosomal recessive, or X-linked pattern. Due to genetic, allelic, and phenotypic heterogeneity, the diagnosis of RP patients can be a challenging task. Until now, mutations in at least 75 genes have been linked to nonsyndromic RP (RetNet, https://sph.uth.edu/retnet/; 10 January 2023). Molecular diagnosis is unquestionably crucial for establishing a correct clinical diagnosis in patients presenting with RP, and it is also necessary for estimating the risk of recurrence and providing reliable genetic counseling to families. Finally, it will form the basis for gene-specific therapy in the future.

A significant improvement of diagnostic yields has been gained in recent years with the use of next generation sequencing-based approaches, which are highly efficient for testing causal genetic variants and helpful in elucidating many previously unexplained causes of RP.

In this study, we implemented a WES re-diagnosis in a cohort of Polish patients affected with nonsyndromic RP, in whom a NGS diagnostic panel performed several years ago failed to identify any potentially causative variants.

## 2. Materials and Methods

### 2.1. Clinical Examination

Nineteen probands, together with 8 affected siblings with an initial diagnosis of nonsyndromic RP and 14 unaffected family members, were recruited from 19 unrelated Polish families. Within the cohort, 8 cases were familial, and 11 were sporadic. The patients underwent genetic counseling, coupled with a detailed medical history analysis. Ophthalmologic examinations, including measurement of central visual acuity, an eye fundus examination, electroretinography (ERG), and optical coherence tomography (OCT), were performed. All participants or guardians provided written informed consent and the study was approved by the Bioethics Committee at Poznan University of Medical Sciences in compliance with good clinical practice (GCP) and Polish law. All study protocols adhered to the tenets of the Declaration of Helsinki.

### 2.2. Molecular Analysis

Blood samples from 19 probands and 22 tested family members were obtained for genetic analysis. Genomic DNA was extracted from the peripheral blood lymphocytes using a MagCore^®^ HF16 Automated Nucleic Acid Extractor and Genomic DNA Large Volume Whole Blood Kit (RBC Bioscience Corp., New Taipei City, Taiwan).

#### 2.2.1. Targeted Next-Generation Sequencing

Diagnostic NGS panels associated with autosomal dominant or autosomal recessive retinitis pigmentosa, encompassing 26 and 56 genes (Appendix A) respectively, were first performed in the probands, according to the expected type of inheritance. Targeted NGS of Retinitis Pigmentosa was performed at Asper Biogene (Republic of Estonia). For details see the Supplementary Material. The variant classification was established based on American College of Medical Genetics (ACMG) guidelines [4]. 

#### 2.2.2. Whole-Exome Sequencing

DNA samples of patients from S6 to S19 were subjected to exome capture and high-throughput sequencing. Target enrichment was performed using a Twist Human Core Exome Kit (Twist Bioscience, San Francisco, CA, USA), and paired-end sequencing (2 × 100 bp) was carried out using Illumina NovaSeq 6000 (Illumina). A variant-discovery pipeline was built on the basis of GATK Best Practices. Variants (hg19 reference) were annotated using ANNOVAR [5] (all non-commercially available component databases as of 16 December 2020), Ensembl/VEP [6] 102.0, CADD [7], and v1.6, Exomiser v12.1.0 [8]. Frequent (AF > 0.001 in the component population bases) and/or benign variants (according to ClinVar; as of 16 December 2020) were discarded at the filtering stage. For phenotypic scoring, Phen2Gene [9] was used to rescore Exomiser results, based on Human Phenotype Ontology term HP:0000510 [10]. CNVs were analyzed manually based on a priority list, according to Phen2Gene scores.

#### 2.2.3. Sanger Sequencing

The candidate variants identified in each patient were validated using PCR and Sanger sequencing. Furthermore, Sanger sequencing of the appropriate gene fragments was performed, in order to carry out segregation analysis and establish the inheritance mode of the altered alleles in selected families, given the availability of the relatives’ DNA samples. PCR reactions and purification of PCR products were carried out following standard protocols. 

#### 2.2.4. Quantitative Real-Time Polymerase Chain Reaction (qPCR)

Quantitative real-time polymerase chain reaction (qPCR) was performed to validate the CNVs and to narrow down their genomic coordinates. A previously described protocol was followed [11]. For details see the Appendix A. For primer sequences see Appendix A.

#### 2.2.5. Breakpoint Sequencing

A series of qPCR, long-range PCR, and standard PCR amplification coupled with Sanger sequencing were utilized to establish the exact genomic coordinates of the aberrations using primers specific to the DNA fragments overlapping the 5′ and 3′ ends of the CNVs. qPCR was performed as described in the previous paragraph (additionally for details see the Appendix A). The long-range PCR was performed using Ranger Mix (Bioline) following the producer’s protocol. For primer sequences see Appendix A. 

## 3. Results

### 3.1. Clinical Features

In the current study, we present 19 patients diagnosed with nonsyndromic RP. All probands displayed classic RP features, with loss of night vision as the first symptom of the disease, accompanied by a progressing narrowing of the visual field in the following years. The fundus examination showed retinal bone-spicule pigmentation (except for patient S16) and abnormal pallor of the optic disc in all patients. Cataract was observed in patients S3, S6, S7, S11, S12, S15, and S16. The ophthalmological findings of the patients are summarized in Table 1. The clinical features of patient S15 are presented in Figure 1. 

### 3.2. Molecular Results

To search for variants that could be responsible for the patients’ phenotypes, we performed targeted NGS sequencing prior to this study. The NGS panel type was selected based on the expected inheritance mode for each patient individually. In each family, an analysis was performed on the index patient. Previously reported and novel variants identified in RP patients are listed in Table 2. Targeted NGS allowed confirmation of the molecular background of the disease in 5 out of 19 patients. Among them, five novel and two previously reported variants were identified in the following genes: *SPATA7*, *CERKL*, *PRPF8*, and *PRPF31* (Table 2).

Fourteen families (from F6 to F19) who remained unsolved despite targeted NGS were subjected to WES. WES revealed potentially causative variants in RP-related genes in 12 out of 14 families, including 8 novel and 10 previously reported variants involving *BBS2*, *USH2A*, *RPGR*, *SNRNP200*, *EYS*, *RGR*, *PRPF31*, *CEP290*, *NR2E3*, and *PRPF4* (Table 2). The analysis targeting copy number variants (CNVs) using WES data allowed for the identification of three novel heterozygous CNVs in patients S9, S12, and S15 (Appendix A). We utilized qPCR and Sanger sequencing to narrow down the genomic coordinates of all aforementioned alterations and were able to show the span and orientation of the CNVs detected in probands S5, S9, and S15 (Table 2, Appendix A). In family S9, the identified deletion of 63,759 bp (chr1:216,259,404-216,323,160; hg19) encompassing exons 22–24 of the *USH2A* gene (Appendix A) was combined with a novel splicing variant c.3316 + 1G > T (Appendix A). The size of the duplication identified in proband S12 was narrowed down using qPCR and its maximal size was 30,101 bp (chr6:65,134,995-65,165,095; hg19) and minimal size was 29,307 bp (chr6:65,135,261-65,164,567; hg19) (Appendix A). Patient S12 carried a heterozygous duplication of exons 27 and 28 and portion of the adjacent introns of the *EYS* gene, as well as a missense variant c.7654del (p.Val2552Ter) on the other allele (Figure 2 and Appendix A).

Deletions of exons 5, 6, and partly 7 in the *PRPF31* gene, 1543 bp in size, were identified in familial case S15 (Figure 2) (chr19:54,625,718-54,627,260, with a ATTATCATT insertion at the breakpoint; hg19) (Appendix A). Unfortunately, the patient’s family members did not consent to participate in genetic testing. We were not successful in identifying the molecular basis of the disease in patients S10 and S16. Due to the consanguinity of the parents of patient S10, we assumed autosomal recessive inheritance of the variant but did not identify a potentially pathogenic variant in the *RP* genes. Based on the analysis of the pedigree of family F16, an autosomal dominant mode of inheritance of the disease was expected; however, the WES analysis failed to uncover a dominant causal variant. The only identified variant was a heterozygous deletion in the *RP1L1* gene, responsible for the autosomal recessive form of the disease, which was not compatible with the history of RP in this family (Figure 2).

None of the novel variants were present in the gnomAD SVs (v2.1) (Genome Aggregation Database) and the NHLBI Exome Variant Server (EVS), and therefore, based on the ACMG guidelines, they were assessed as pathogenic or likely pathogenic variants. With the use of the two NGS diagnostic methods applied, it was possible to identify the potentially causative gene variants in 17 out of 19 patients included in this study. We identified 25 causal gene variants, of which 13 were novel. The segregation of the variants identified in familial cases is presented in Figure 2.

## 4. Discussion

In recent years, an increasing number of genetic determinants of retinitis pigmentosa has necessitated a drive towards the use of next-generation sequencing [12,13]. While common variants in 75 well-characterized genes have provided an explanation in some nonsyndromic cases, phenotypic variability, the overlap with other retinal disorders, and uncommon causal variants require following up with whole-exome/genome sequencing. In this context, decreasing costs and the improved quality of results, together with increasing coverage of problematic regions of the human genome facilitate a more thorough analysis [2]. For complex phenotypes with multiple underlying causes, such as RP, successful identification of causative structural variations and regulatory changes requires prioritization of clinically relevant variants coupled with phenotype ontology [13]. 

In this report, next generation sequencing methods revealed the coexistence of causal variants affecting distinct RP genes in 17 out of 19 RP families. Previously, using targeted NGS analysis, we had detected variants associated with the disease in only five patients from this group, whereas with the use of WES, we established the molecular background of the disease in another 12 RP patients. In the group presented, it was possible to determine the genetic basis of the disease with very high efficiency, which reached 89%. The overall diagnostic yield of 89% was distinctively higher than that presented in previous studies, where the estimated detection rate ranged from 50 to 70% [14,15,16,17]. The choice of the NGS panel in the past was carried out based on the type of inheritance of the disease expected. However, this approach may be ineffective in sporadic cases of RP, which was noticeable in the presented cohort. One issue was a different type of inheritance being identified after WES than the one suggested by the previously selected NGS panel. This instance was observed in sporadic patients S6, S7, S11, S13, S14, and the familial case S8. Furthermore, in patients S9 and S12, heterozygous variants in recessive genes were identified in the NGS panel, while the second causal variant for each patient was only found by WES. CNVs were the second alterations in both patients (Table 2). It is worth mentioning that, due to a lower coverage of targeted regions and restricted bioinformatic capabilities, the NGS panel analysis performed several years ago did not routinely include a pipeline for detection of CNVs. Patients S15 and S17 harbored mutations in genes that were not captured by gene panels and for the aforementioned reasons were only detected by means of WES. 

In the presented RP cohort, in nine patients, the disease is inherited in autosomal recessive mode and in eight patients in autosomal dominant mode, while in two patients it was X chromosome-linked. The expression of RP inherited in an autosomal dominant mode is the least severe, associated with a delayed onset of symptoms and slower progression of the disease, which is in line with observations in our group of patients. The clinical features are relatively more severe in patients with X-linked variants compared to patients with other types of inheritance [18]. Affected males usually experience an early onset of symptoms and present with rapid progression, while female carriers can show great phenotypic variability, from asymptomatic to significant visual and retinal impairment, and present the first symptoms much later [19]. family F8, although the first symptoms in the affected patient S8 appeared relatively late (at 15 years of age) compared to other patients with X-linked RP, currently he presents with severely decreased visual acuity and an extremely restricted visual field (Table 2). The female carrier from the same family (I.2) experienced decreased visual acuity at the age of 30, and from the age of 60 progressive narrowing of the visual field and night blindness have also been observed. The three remaining carriers in the family (III.2, III.3, and III.4) have shown no symptoms of RP to date (Figure 2). As reported in the literature, disease expression in female carriers can be correlated with X chromosome methylation status. Even though numerous studies did not show a strong correlation between X-inactivation (XCI) and the carriers’ phenotype, retinal XCI ratios may not be accurately reflected in the most commonly tested blood DNA [20]. Variable expression of the disease in female carriers with skewed X-inactivation and overexpression of the mutant gene in the retina remains a feasible explanation. 

Furthermore, in this study, we reported a case of pseudodominantly inherited autosomal recessive RP in the family F18 with co-segregating deleterious variants in the *NR2E3* gene. Based on the family F18 pedigree, RP was inherited in an apparent autosomal dominant pattern (Figure 2). Targeted NGS revealed the known pathogenic homozygous variant c.481delA (p.Thr161HisfsTer18) in one patient from this family (III.3), suggesting the diagnosis of an autosomal recessive form of the disease, which was in contradiction with the observed pedigree. Segregation analysis of the variant in the family confirmed the presence of the homozygous variant in an affected brother (III.2) and indicated the clinically healthy carriers (I.2, II.4, III.1) (Figure 2). Interestingly, we also identified the same but heterozygous alteration in the patient’s affected father (II.3). Therefore, to search for another causative variant in patient S18, WES was conducted. This revealed another c.951delC (p.Thr318fs) alteration and validated the presence of the c.481delA (p.Thr161HisfsTer18) mutation in the *NR2E3* gene in a compound heterozygous state (Appendix A). This finding confirmed the pseudodominant inheritance of RP in family F18. Pseudodominant inheritance of autosomal recessive diseases occurs when an affected individual (homozygous or compound heterozygous) has an unaffected partner, who is a carrier for a pathogenic variant in the same gene, and their offspring is affected by the same autosomal recessive disorder as manifested in the parent. Pseudodominant inheritance has already been documented in retinal disorders, including congenital stationary night blindness (CSNB) with alterations in the *GRM6* gene [21], in patients with RP related to *IMPG2* variants, or in cone-rod dystrophy and Stargardt disease with *ABCA4* pathogenic variants [22,23]. Nevertheless, to our knowledge this is the first report of *NR2E3* pseudodominant inheritance in retinitis pigmentosa. This finding reminds us of the importance of considering the possibility of pseudodominance in apparently autosomal dominant pedigrees of RP. It also indicates the crucial role of molecular analysis, which in some cases is the only means of correctly classifying the mode of inheritance as pseudodominant.

A heterozygous deletion encompassing the entire sequence of the *PRPF31* gene was detected by targeted NGS and confirmed by qPCR in patient S5. Breakpoint analysis determined the size of the deletion as 33,998 bp (chr19:54,602,534-54,636,531; hg19). In addition to the *PRPF31* gene, the detected deletion encompasses three upstream coding genes: *TFPT*, *NDUFA3*, and a portion of *OSCAR*. The presence of the deletion was also confirmed in the patient’s affected father using qPCR (Figure 2 and Appendix A). *PRPF31* mutations are the second most common genetic cause of adRP in most populations [24]. Deletions encompassing the entire *PRPF31* and upstream genes have previously been reported in RP patients [25,26,27]. 

In this study, we did not find disease-causing alterations in two patients (S10, S16). Possible reasons for this could be that the variants are in the noncoding regions of corresponding genes that cannot be captured by WES analysis or that they are difficult-to-identify gross deletions, insertions, or complex rearrangements. Whole-genome sequencing may be a comprehensive alternative method that could resolve these problems.

The considerable advancements in high-throughput sequencing observed in recent years have made molecular diagnosis possible for those patients in whom prior testing failed to identify the genetic basis of disease. Therefore, it is important for previously undiagnosed patients to undergo a high-throughput sequencing again. Due to the higher sequencing depth and broader target enrichment than in the NGS panels, when coupled with suitable bioinformatic analysis, WES can be sufficient to accurately identify disease-causing variants in most RP patients. In conclusion, our study confirmed the efficiency and clinical utility of re-diagnosis with whole-exome sequencing in molecularly undiagnosed patients with RP. It also showed that WES is suitable and efficient for the molecular screening of RP patients. Performing WES enabled us to establish a final diagnosis in most RP patients presented in this study and made it feasible to assess the patients’ prognosis.

## Figures and Tables

**Figure 1 diagnostics-13-00730-f001:**
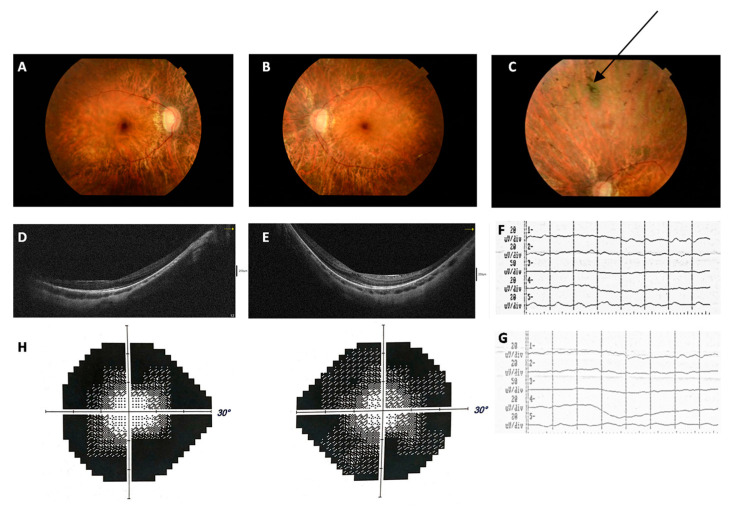
Clinical features of patient S15 with RP. Fundus photography of the right eye (**A**) and left eye (**B**), showing typical features of RP including optic disc pallor, attenuated retinal vessels, and mid-peripheral bone-spicule pigmentation (indicated by an arrow) (**C**). (**D**,**E**) Optical coherence tomography (OCT) showed thinning of the retina and loss of photoreceptors out of the fovea. In the RE visible single cystic changes in the foveola and fine epiretinal membrane in the temporal part of the macula. (**F**,**G**) Full-field electroretinogram (ERG) demonstrated residual rod and cone response amplitude. (**H**) Concentric narrowing (to approx. 10°) of the visual field of both eyes.

**Figure 2 diagnostics-13-00730-f002:**
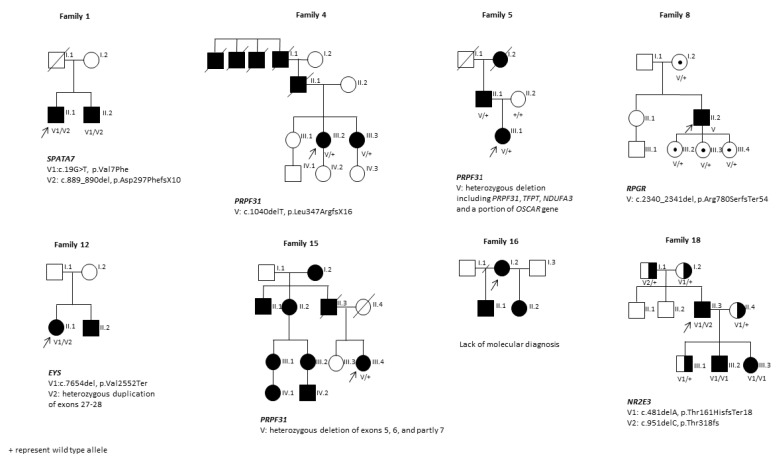
Pedigrees and segregation of the variants associated with the disease identified in familial RP cases. Genotypes are provided for all subjects available for molecular analysis. Identified pathogenic variants are placed below the pedigree. Wild-type variants are indicated with +, while disease-causing variants are indicated with V1 and V2.

**Table 1 diagnostics-13-00730-t001:** Clinical symptoms of patients with RP.

Patient ID/Family	Age Ranges/ Gender	Family History	Night Blindness/Age	Anterior Segment	Optic Nerve Pallor	Arteriolar Attenuation	Macula	Peripheral Retina	BCVA RE/LE	VF Restriction	ERG Results
**S1/F1**	20s/M	+	+/ND	−	+	+	without reflex	bone-spicule	0.2/0.1	ND	extinguished
**S2/F2**	30s/M	−	+/16	−	+	+	without reflex, bull’s eye maculopathy	bone-spicule, “salt and pepper”	0.7/0.8	restricted	extinguished
**S3/F3**	40s/F	−	+/7	cataract	+	+	normal	bone-spicule	0.9/0.9	5–10°	extinguished
**S4/F4**	40s/F	+	+/childhood	−	+	+	without reflex, mild dystrophic changes	bone-spicule	1.0/1.0	3–5°	ND
**S5/F5**	10s/F	+	+/12	−	+	+	without reflex	bone-spicule	ND	30°	scotopic extinguished
**S6/F6**	40s/M	−	+/20	cataract	+	+	without reflex	bone-spicule	0.5/0.5	restricted	extinguished
**S7/F7**	60s/F	−	+/childhood	cataract	+	+	normal	bone-spicule	1.0/0.6	<10°	ND
**S8/F8**	40s/M	+	+/15	−	+	+	hyperpigmentation	bone-spicule	0.2/0.1	<10°	scotopic extinguished, fotopic <10%
**S9/F9**	30s/F	−	+/21	−	+	+	without reflex	bone-spicule	ND	10°	extinguished
**S10/F10**	30s/M	−	+/-	−	+	+	normal	bone-spicule	1.0/1.0	15°	residual
**S11/F11**	50s/M	−	+/30	cataract	+	+	Hyperpigmentation, ERM	bone-spicule	0.15/0.1	10°	diminished
**S12/F12**	40s/F	+	+/15	cataract	+	+	without reflex	bone-spicule	0.1/0.1	<10°	residual
**S13/F13**	40s/M	−	+/40	−	+	−	CME	hyperpigmentation	1.0/1.0	10°	residual
**S14/F14**	30s/M	−	+/childhood	−	+	+	macular degeneration, CME	bone-spicule	0.6/0.8	10–15°	residual
**S15/F15**	30s/F	+	+/6	cataract	+	+	without reflex	bone-spicule	0.2/0.2	10°	residual
**S16/F16**	10s/F	+	+/childhood	cataract	+	+	normal	no changes	1.0/1.0	30°	ND
**S17/F17**	20s/M	−	+/7	−	+	+	without reflex	bone-spicule	0.2/0.2	10°	residual
**S18/F18**	40s/M	+	+/early childhood	−	+	+	normal	bone-spicule	1.0/1.0	restricted	residual
**S19/F19**	30s/F	−	+/20	−	+	+	without reflex	bone-spicule	0.9/0.7	<10°	diminished

BCVA—best-corrected visual acuity. RE—right eye. LE—left eye. M—male. F—female. VF—visual field. ND—no data.

**Table 2 diagnostics-13-00730-t002:** Pathogenic DNA variants identified in RP patients.

Patient/Family	Mode of Inheritance	Gene	Transcript	Variant Classification		Pathogenicity Prediction in Protein Level			ACMG Classification	Molecular Method of Searching the Variants
				Nucleotide	Protein	SIFT	PolyPhen-2	CADD		
**S1/F1**	AR	*SPATA7*	NM_018418.5	c.19G>T ***c.889_890del***	p.Val7Phe ***p.Asp297PhefsX10***	D -	PD -	- -	Pathogenic Pathogenic	NGS panel
**S2/F2**	AR	*CERKL*	NM_201548.5	** *c.397_401del* ** ** *c.1222G>T* **	** *p.Leu133GlufsX5* ** ** *p.Gly408X* **	- -	- -	- -	Pathogenic Likely pathogenic	NGS panel
**S3/F3**	AD	*PRPF8*	NM_006445.4	** *c.6974_6985del* **	** *p.Val2325_Ala2328del* **	-	-	-	Likely pathogenic	NGS panel
**S4/F4**	AD	*PRPF31*	NM_015629.4	** *c.1040delT* **	** *p.Leu347ArgfsX16* **	-	-	-	Pathogenic	NGS panel
**S5/F5**	AD	*PRPF31*	NM_015629.4	heterozygous deletion chr19:54,606,405-54,637,153		-	-	-	-	NGS panel
**S6/F6**	AR	*BBS2*	NM_031885.4	c.815G>A c.653G>A	p.Arg272Gln p.Gly218Asp	D D	PD PD	D LD	Pathogenic Pathogenic	WES
**S7/F7**	AR	*USH2A*	NM_206933.3	c.14926G>A ***c.292A>C***	p.Gly4976Ser ***p.Thr98Pro***	D B	PD PD	LD -	Pathogenic Pathogenic	WES
**S8/F8**	X-linked	*RPGR*	NM_001034853.2	** *c.2340_2341del* **	** *p.Arg780SerfsTer54* **	-	-	LD	Pathogenic	WES
**S9/F9**	AR	*USH2A*	NM_206933.3	** *c.3316+1G>T* ** * **heterozygous deletion** * * **c.(4627+1_4628-1)_(4978+1_4979-1)del** *	- -	- -	- -	- -	Pathogenic -	WES
**S11/F11**	AD	*SNRNP200*	NM_014014.5	** *c.1671+19C>A* **	-	-	-	-	-	WES
**S12/F12**	AR	*EYS*	NM_001292009.1	c.7654del ***heterozygous duplication*** ***c.(2732+1_2733-1)_(6078+1_6079-1)dup***	p.Val2552Ter -	- -	- -	D -	Pathogenic -	WES
**S13/F13**	AD	*RGR*	NM_002921	c.806A>G	p.Tyr269Cys	D	PD	LD	Pathogenic	WES
**S14/F14**	X-linked	*RPGR*	NM_001034853.2	** *c.1070G>A* **	** *p.Gly357Asp* **	LD	PD	LD	Likely pathogenic	WES
**S15/F15**	AD	*PRPF31*	NM_015629.4	** *heterozygous deletion* ** * **c.(420+1_421-1)_(697+1_698-1)del** *	-	-	-	-	-	WES
**S17/F17**	AR	*CEP290*	NM_025114.4	c.1984C>T c.223A>G	p.Gln662X p.Lys75Glu	- B	- LD	D D	Pathogenic Uncertain significance	WES
**S18/F18**	AR	*NR2E3*	NM_014249.4	c.481delA c.951delC	p.Thr161HisfsTer18 p.Thr318fs	- -	- -	- -	Pathogenic Pathogenic	WES
**S19/F19**	AD	*PRPF4*	NM_004697.4	c.1331C>T	p.Thr444Ile	D	D	LD	Benign	WES

WES—whole-exome sequencing, D—damaging, PD—probably damaging, LD—likely damaging, B—benign. American College of Medical Genetics (ACMG) classification was obtained through the Varsome online available tool. Human Genome Variation Society (HGVS) nomenclature have been used. New variants identified in this study are in bold.

## Data Availability

The data used in this study are available from the corresponding author upon request.

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
