# Peer review of "Molecular Re-Diagnosis with Whole-Exome Sequencing Increases the Diagnostic Yield in Patients with Non-Syndromic Retinitis Pigmentosa"

_diagnostics, 2023, doi:10.3390/diagnostics13040730_

Round 1

Reviewer 1 Report

This is a well - written manuscript. The study reports the efficiency and clinical utility of 25 re-diagnosis with WES in molecularly undiagnosed RP patients. 

I do not understand why authors state that data availability is not applicable. Data should be available for this kind of studies.  

Author Response

Dear Editor,

Thank you for considering our manuscript entitledMolecular re-diagnosis with Whole-Exome Sequencing increases the diagnostic yield in patients with non-syndromic Retinitis Pigmentosa” for publication in the Diagnostics. We are very grateful to the reviewers for their valuable evaluation of the paper and suggestions.  We have made the recommended changes, as detailed in the attached response  to the reviewers. We hope that our revised manuscript will reflect an adequate response to the reviewers comments.

Reviewer 1: I do not understand why authors state that data availability is not applicable. Data should be available for this kind of studies.                                   Response: Data availability statement has been changed in the manuscript.  The data used in this study are available from the corresponding author upon request.

Reviewer 2 Report

In this manuscript, Nineteen unrelated Polish probands clinically diagnosed with nonsyndromic RP were recruited for this study. Targeted NGS identified the molecular background in only 5 out of 19 patients. The authors used Whole Exome Sequencing (WES) to identify potential pathogenic gene variants in molecularly undiagnosed 14 unrelated Polish RP patients. WES revealed potentially causative variants in RP-related genes in another 12 patients. In conclusion, NGS methods revealed the coexistence of causal variants affecting distinct RP genes in 17 out of 19 RP families with a very high efficiency of 89%. The authors' study supported the importance of repeating high-throughput sequencing analysis in patients whose previously performed NGS did not reveal any pathogenic variants.

The study confirms the efficiency and clinical utility of re-diagnosis with WES in molecularly undiagnosed RP patients. This manuscript is compelling and interesting, the data is clear, and the discussions are adequate.

To make the manuscript more profound, I would like to give three suggestions:

1.      Please remake Table 1 and Table 2 to make them more beautiful.

2.      Both environmental factors and gene mutations can contribute to diseases. The patients who were not found disease-causing alterations may have been caused by environmental factors. Please discuss the contribution of environmental factors to the retinitis pigmentosa.

3.      Please add the pedigree of family 6 and family 17 in Figure 2.

Author Response

Dear Editor,

Thank you for considering our manuscript entitledMolecular re-diagnosis with Whole-Exome Sequencing increases the diagnostic yield in patients with non-syndromic Retinitis Pigmentosa” for publication in the Diagnostics. We are very grateful to the reviewers for their valuable evaluation of the paper and suggestions.  We have made the recommended changes, as detailed in the attached response  to the reviewers.

  1. Please remake Table 1 and Table 2 to make them more beautiful.  Response: The tables have been corrected.
  2. Both environmental factors and gene mutations can contribute to diseases. The patients who were not found disease-causing alterations may have been caused by environmental factors. Please discuss the contribution of environmental factors to the retinitis pigmentosa.                          Response: 

    Retinitis pigmentosa is an inherited disorder, and therefore not caused by injury, infection or any other external or environmental factors, except for the genetic factor. However there are many risk factors that may have effect on the progression of RP, such as smoking, diet, physical activity, inflammantory factors, intraocular inflammation or oxidative stress. In addition, there are diseases resembling RP, such as autoimmune, paraneoplastic or inflammatory diseases, which, however, are not hereditary forms of RP. In our study, we did not find the genetic cause of the disease in two RP patients (S10, S16). We believe that this is due to the presence of causal variants in non-coding sequences of corresponding genes   that cannot be captured by Whole exome sequencing  analysis or they are difficult to identify deletions, insertions, or complex rearrangements. Since ophthalmological examinations of patients S10 and S16 clearly indicate a hereditary form of RP.

  3. Please add the pedigree of family 6 and family 17 in Figure 2.              Response: Figure 2 shows only familial cases of RP patients, patients from family 6 and 17 are sporadic cases and for this reason were not included in this figure.